# Prognostic Significance of Organ-Specific Metastases in Patients with Metastatic Upper Tract Urothelial Carcinoma

**DOI:** 10.3390/jcm11185310

**Published:** 2022-09-09

**Authors:** Antonio Tufano, Nadia Cordua, Valerio Nardone, Raffaele Ranavolo, Rocco Simone Flammia, Federica D’Antonio, Federica Borea, Umberto Anceschi, Costantino Leonardo, Andrea Morrione, Antonio Giordano

**Affiliations:** 1Department of Urology, University Sapienza, 00185 Rome, Italy; 2Sbarro Institute for Cancer Research and Molecular Medicine, Center for Biotechnology, Department of Biology, College of Science and Technology, Temple University, Philadelphia, PA 19122, USA; 3Department of Biomedical Sciences, Humanitas University, 20072 Pieve Emanuele, Italy; 4Department of Precision Medicine, University of Campania “L. Vanvitelli”, 80138 Naples, Italy; 5Urology Unit, AORN Ospedali dei Colli-Monaldi Hospital, 80131 Naples, Italy; 6Department of Urology, Regina Elena National Cancer Institute, 00144 Rome, Italy; 7Department of Medical Biotechnology, University of Siena, 53100 Siena, Italy

**Keywords:** metastatic upper tract urothelial carcinoma, metastatic organ, surveillance, Epidemiology and End Results, prognosis

## Abstract

Background: Existing data on metastatic upper tract urothelial carcinoma (mUTUC) are limited. In this study, we investigated the prognostic value of site-specific metastases in patients with mUTUC and its association with survival outcomes. Methods: We retrospectively collected data from the Surveillance, Epidemiology and End Results (SEER) database between 2004 and 2016. Kaplan–Meier analysis with a log-rank test was used for survival comparisons. Multivariate Cox regression was employed to predict overall survival (OS) and cancer-specific survival (CSS). Results: 633 patients were selected in this study cohort. The median follow-up was 6 months (IQR 2–13) and a total of 584 (92.3%) deaths were recorded. Within the population presenting with a single metastatic organ site, the most common metastatic sites were distant lymph nodes, accounting for 36%, followed by lung, bone and liver metastases, accounting for 26%, 22.8% and 16.2%, respectively. In patients with a single metastatic organ site, the Kaplan–Meier curves showed significantly worse OS for patients with liver metastases vs. patients presenting with metastases in a distant lymph node (*p* < 0.001), bone (*p* = 0.023) or lung (*p* = 0.026). When analyzing CSS, statistically significant differences were detectable only between patients presenting with liver metastases vs. distant lymph node metastases (*p* < 0.001). Multivariate analyses showed that the presence of liver (OS: HR = 1.732, 95% CI = 1.234–2.430, *p* < 0.001; CSS: HR = 1.531, 95% CI = 1.062–2.207, *p* = 0.022) or multiple metastatic organ sites (OS: HR = 1.425, 95% CI = 1.159–1.753, *p* < 0.001; CSS: HR = 1.417, 95% CI = 1.141–1.760, *p* = 0.002) was an independent predictor of poor survival. Additionally, survival benefits were found in patients undergoing radical nephroureterectomy (RNU) (OS: HR = 0.675, 95% CI = 0.514–0.886, *p* = 0.005; CSS: HR = 0.671, 95% CI = 0.505–0.891, *p* = 0.006) and chemotherapy (CHT) (OS: HR = 0.405, 95% CI = 0.313–0.523, *p* < 0.001; CSS: HR = 0.435, 95% CI = 0.333–0.570, *p* < 0.001). Conclusions: A distant lymph node was the most common site of single-organ metastases for mUTUC. Patients with liver metastases and patients with multiple organ metastases exhibited worse survival outcomes. Lastly, CHT administration and RNU were revealed to be predictors of better survival outcomes in the mUTUC cohort.

## 1. Introduction

Upper tract urothelial carcinoma (UTUC) is a rare malignancy that accounts for 5% to 10% of all urothelial cancers. The annual incidence of UTUC is typically estimated at 2 per 100,000 people in Western countries, with a peak in individuals aged from 70 to 90 years [1].

In patients with a diagnosis of UTUC, the 3-year overall survival (OS) rate is around 75% [2]. However, when considering metastatic UTUC (mUTUC) patients, the 3-year OS rate does not exceed 10% [3].

Recently, the prognostic role of metastasis at distant organs was investigated in multiple types of cancers [4,5,6]. To date, data on metastatic urothelium carcinoma are mainly based on studies on urothelial bladder cancer (UBC). For example, Shou et al. reported that the presence of liver metastasis was an independent predictor of OS when compared with other metastatic organ sites in metastatic bladder cancer (mBCa) [7]. Moreover, Dong et al. suggested that patients with mBCa exhibiting more than one metastatic site presented with unfavorable survival outcomes. Based on these results, we hypothesized that organ-specific metastases might also play a role in the prognosis of mUTUC [8].

Hence, our purpose was to retrospectively investigate the impact of organ-specific mUTUC patients on prognostic survival outcomes.

## 2. Materials and Methods

### 2.1. Database and Patient Selection

In January 2022, we interrogated the Surveillance, Epidemiology and End Results (SEER) database and identified patients with mUTUC from 2004 to 2016 according to the following inclusion criteria: (1) aged ≥ 18 years; (2) UTUC with distant metastasis as the first primary malignancy; (3) pathologically confirmed mUTUC of the renal pelvis, ureter or ureter orifice (International Classification of Disease for Oncology (ICD-O) site codes C65.9, C66.9 and C67.6, respectively); and (4) enrolled patients should have confirmed information on metastases in their bone, liver, lung, brain or distant lymph node. We excluded any metastatic pattern presented in less than 20 patients. The inclusion process is presented in Appendix A. SEER*Stat software (SEER*Stat 8.2.3) was used to extract data.

### 2.2. Study Outcomes

Overall survival (OS) and cancer-specific survival (CSS) were the two major outcomes of this study. OS was defined as the interval of time from the UTUC diagnosis to death for any cause. CSS was defined as the interval from the date of UTUC diagnosis to death due to the tumor.

### 2.3. Statistical Analysis

Demographic factors were reported as frequency and proportion. The Mantel–Cox log-rank test was applied to compare the Kaplan–Meier survival curves for OS and CSS. Univariable and multivariable Cox proportional models with hazard ratios (HRs) and 95% confidence intervals (95% CIs) addressing both OS and CSS were performed. Two-sided *p*-values < 0.05 were considered significant. Statistical analysis was carried out using the Statistical Package for Social Sciences (SPSS) software v.26.0 (IBM Corp, Armonk, NY, USA).

## 3. Results

### 3.1. Demographic and Clinical Characteristics

The demographic characteristics of the included patients are shown in Table 1. A total of 633 mUTUC patients were identified after applying the inclusion/exclusion criteria. The median age at diagnosis was 72 years (IQR 63–79). mUTUC patients were more frequently males (57.8%) and ethnically white (83.6%). According to the AJCC Sixth edition TNM stage, 322 (50.9%) of the patients harbored a T-stage ≥ 2 and 431 (68.1%) had a positive N-stage. Furthermore, 354 (55.9%) mUTUC patients received chemotherapy (CHT) and 247 (39%) underwent radical nephroureterectomy (RNU). The median follow-up period was 6 months (IQR 2–13) and a total of 584 (92.3%) deaths were recorded.

### 3.2. Metastatic Patterns

Information on the metastatic organs is summarized in Table 1. Overall, 65.9% (*n* = 417) of the included population presented with a single metastatic organ site. Of those, the distribution showed that 36% (*n* = 147) were diagnosed with distant lymph nodes metastases, followed by lung, bone and liver metastases accounting for 26.0% (*n* = 106), 21.8% (*n* = 89) and 16.2% (*n* = 66), respectively. Moreover, 34.1% (*n* = 216) of patients presented with two or more metastatic organ sites.

### 3.3. Impact of Metastatic Sites and Treatment Scheme on Survival Outcomes

First, the OS and CSS were analyzed according to single- vs. multiple-organ metastases. Our results revealed that patients with a single metastatic organ site had significantly better outcomes for both OS (*p* < 0.001) and CSS (*p* < 0.001) (Figure 1a,b).

Second, we focused on the OS and CSS in patients with a single metastatic organ site. Here, statistically significant differences were recorded between the Kaplan–Meier curves, showing worse outcomes in terms of OS for patients presenting liver metastases when compared with patients presenting distant lymph node (*p* < 0.001), bone (*p* < 0.023) and lung (*p* < 0.026) metastases (Figure 1a). Additionally, a difference was recorded between patients with lung vs. bone metastases (*p* = 0.026). When analyzing the CSS, a statistically significant difference was found only between liver vs. distant lymph node metastases (*p* < 0.001) (Figure 2a,b).

Third, we analyzed survival outcomes stratified by the treatment scheme (yes-CHT vs. no-CHT groups and yes-RNU vs. no-RNU groups) in patients harboring either single- or multiple metastatic organ sites.

Here, statistically significant differences were recorded, showing better survival outcomes in the yes-CHT vs. no-CHT group for both single- and multiple metastatic organ sites (OS and CSS: each *p* < 0.001). Additionally, in the subset of yes-CHT patients, better outcomes were recorded for single vs. multiple metastatic organ sites (OS: *p* = 0.003, CSS *p* = 0.011) (Figure 3a,b).

When focusing on RNU treatment, we observed better survival outcomes in the yes-RNU vs. no-RNU group for single (OS: *p* = 0.022, CSS: *p* = 0.010) and multiple metastatic organ sites (OS and CSS: each *p* < 0.001). Lastly, in the subset of yes-RNU patients, better outcomes were observed in the single- vs. multiple-site groups (OS: *p* = 0.197, CSS *p* = 0.234) (Figure 4a,b).

The parameters age, sex, T-stage, N-stage, surgery of primary site, CHT and number of organ metastases were included in the multivariable Cox regression model (Table 2). Our analysis showed that the subgroup of patients presenting liver metastases had lower OS than patients with distant lymph node metastases (HR = 1.732, 95% CI: 1.234–2.430, *p* < 0.001). Similar results were found for CSS (HR = 1.531, 95% CI: 1.062–2.207, *p* = 0.022). Moreover, when comparing the number of metastatic organ sites (single vs. multiple), significantly longer OS (HR = 1.425, 95% CI: 1.159–1.753, *p* = < 0.001) and CSS (HR = 1.417, 95% CI: 1.141–1.760, *p* = 0.002) were observed in patients with single-organ metastases. Finally, patients undergoing RNU and CHT showed benefit in terms of OS and CSS (RNU—OS: HR = 0.675, 95% CI: 0.514–0.886, *p* = 0.005; CSS: HR = 0.671, 95% CI: 0.505–0.891; *p* = 0.006) (CHT—OS: HR = 0.405, 95% CI: 0.313–0.523, *p* < 0.001; CSS: HR = 0.435, 95% CI: 0.333–0.570, *p* < 0.001).

## 4. Discussion

Previous studies demonstrated that advanced UTUC is aggressive and associated with a poor prognosis. At the time of diagnosis, 50–60% of patients with UTUC present locally advanced disease and up to 25% present with distant metastasis with a median OS of approximately 9 months [9,10,11]. These poor survival outcomes point out the fact that improving the treatment for this disease is still an essential issue and highlight the necessity to identify clinical factors that might improve clinical decision-making. Thus, in this study, we aimed at describing the pattern and frequency of metastatic sites and the prognostic relevance of any specific metastatic site by retrospectively analyzing a large patient cohort using the SEER database.

First, we observed important differences in metastatic sites, where we discovered a high frequency of distant lymph node metastases in patients that presented with a single metastatic organ site. These results are in disagreement with data reported from other mUTUC cohorts, as Chen et al. found that the lung represented the most common metastatic site, accounting for 42.3% of the whole cohort [12]. A possible interpretation of this difference might be that their analysis was limited to mUTUC presenting with a renal pelvis primary.

Second, our analysis focused on OS and CSS. Here, patients presenting with multiple metastases had worse OS and CSS according to Kaplan–Meier curves, which was confirmed using Cox regression analysis (OS: HR = 1.425, CSS: HR = 1.417). This finding is likely supported by the fact that the increase in tumor burden is usually associated with aggressive tumors and, at the same time, limits the possibility of a radical approach that can be used in oligometastatic patients (i.e., surgery or radiotherapy) [13]. Similar to our work, Li et al. analyzed the prognostic factors for OS in patients with mUTUC; the results of their multivariate analysis showed that the number of metastatic organs (1–2 vs. ≥3) outperformed the presence of visceral metastasis [14]. Additionally, Chen et al. reported that patients with ≤2 metastatic sites presented improved survival outcomes when compared with ≥3 metastatic sites [12].

Third, patients presenting with liver metastases had worse OS and CSS, as shown by the Kaplan–Meier curves. Our observations were also confirmed after multivariate adjustments for patient and tumor characteristics, where we identified a statistically significant OS and CSS disadvantage in patients with liver metastases relative to distant lymph node metastases (OS: HR = 1.732, CSS: HR = 1.531). Similarly, Dong et al. demonstrated that patients with liver metastases had worse OS rather than CSS, which provides a possible explanation [8]. Nonetheless, Shou et al. reported that mBCa patients with liver metastasis (SEER database; 2010–2014) exhibited worse OS compared with three other single metastatic organ sites [7]. A possible explanation for these results might be attributed to liver failure and, consequently, an increased overall mortality rate.

Fourth, our analyses focused on CHT administration. CHT is currently the accepted treatment option for advanced UTUC but the majority of data are extrapolated from UBC. In our cohort, CHT was associated with favorable OS (HR = 0.405) and CSS (HR = 0.435). Similar results were reported by Nazzani et al. in a non-surgically treated mUTUC population [15]. In our study, we confirmed the prognostic benefit of CHT in patients with either single or multiple metastatic organ sites. In contrast, Vassilakopoulou et al. demonstrated that adjuvant postoperative CHT does not add any significant benefit with regard to OS in high-risk UTUC patients [16]. However, in this cohort, patients with both locally advanced M0 and M+ were included. Unfortunately, within the SEER database, the proportion of patients that received cisplatin-based CHT vs. carboplatin-based CHT, as well as patients that received adjuvant vs. neoadjuvant CHT, is not specified. Moreover, we were unable to determine the rationale for determining patient exclusion from CHT regimens. However, we registered an administration rate of 55.9%, which is in line with the rates published by Browne et al. based on a large National Cancer Database (NCDB, 2004–2013) [17].

Sixth, we observed a statically significant association between radical nephroureterectomy (RNU) and OS (HR = 0.675) and CSS (HR = 0.671) after multivariable adjustments. Our findings are in agreement with several observational studies that addressed the role of surgery on primary tumor sites for mUTUC, especially in patients fit enough to receive cisplatin-based CHT [18,19].

Taken together, these data described the metastatic patterns of mUTUC and prognosis outcomes. Our results suggested that patients that presented with liver metastases had worse survival outcomes when compared with other metastatic sites. Moreover, patients that presented with metastasis at multiple secondary sites showed a worse prognosis, which was probably related to the greater burden of the disease. Nonetheless, CHT and RNU were found to be protective variables in the multivariate regression survival analysis.

We are aware of the limitations of our work, which should be interpreted in the context of its retrospective and population-based design. The SEER database included only five specific metastatic organs and the information reporting the number of metastases in each organ was not available. In addition, the SEER database does not ascertain either the type or delivery timing of chemotherapy. However, our analysis included a big population of mUTUC and our results can help to better understand and stratify prognoses, as well as justify more aggressive treatment strategies in oligometastatic patients.

## 5. Conclusions

Our analysis showed that a distant lymph node was the most common site of single-organ metastases for mUTUC. Patients with liver metastases and patients with multiple metastatic organ sites exhibited worse survival outcomes. Moreover, CHT administration and RNU were revealed to be predictors of better survival outcomes in the mUTUC cohort. These findings can be useful to stratify prognoses of mUTUC and justify aggressive treatment in oligometastatic patients.

## Figures and Tables

**Figure 1 jcm-11-05310-f001:**
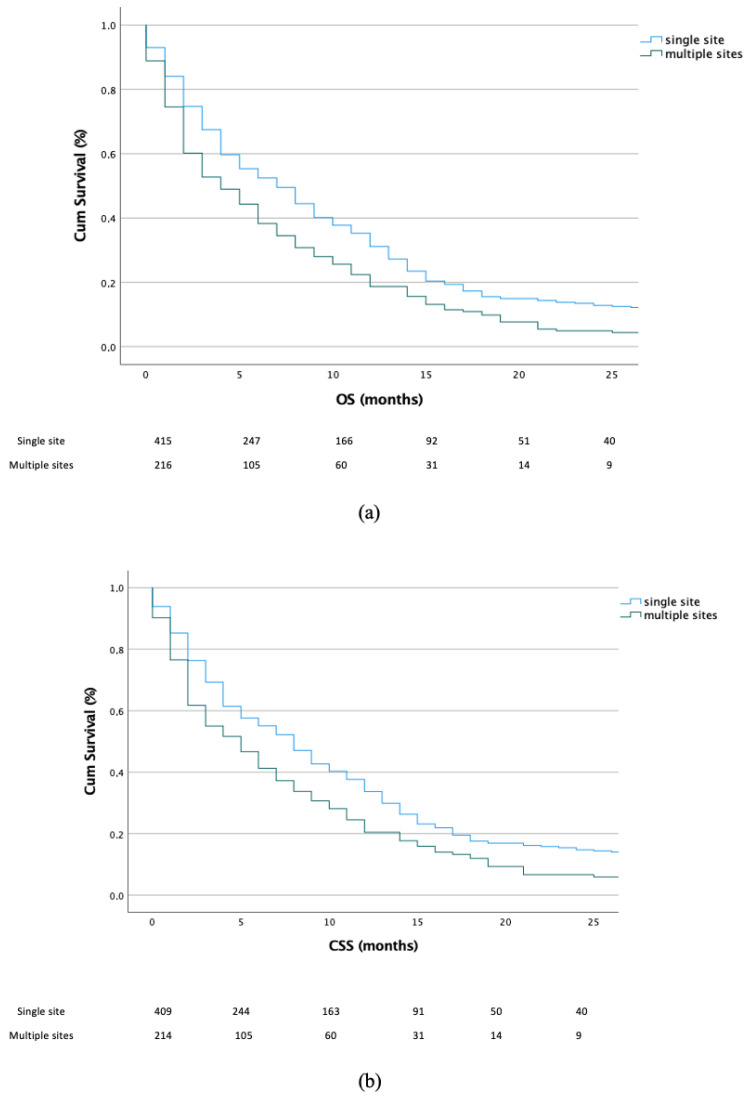
Kaplan–Meier plots depicting the OS (**a**) and CSS (**b**) according to single vs. multiple metastatic organ(s) patients.

**Figure 2 jcm-11-05310-f002:**
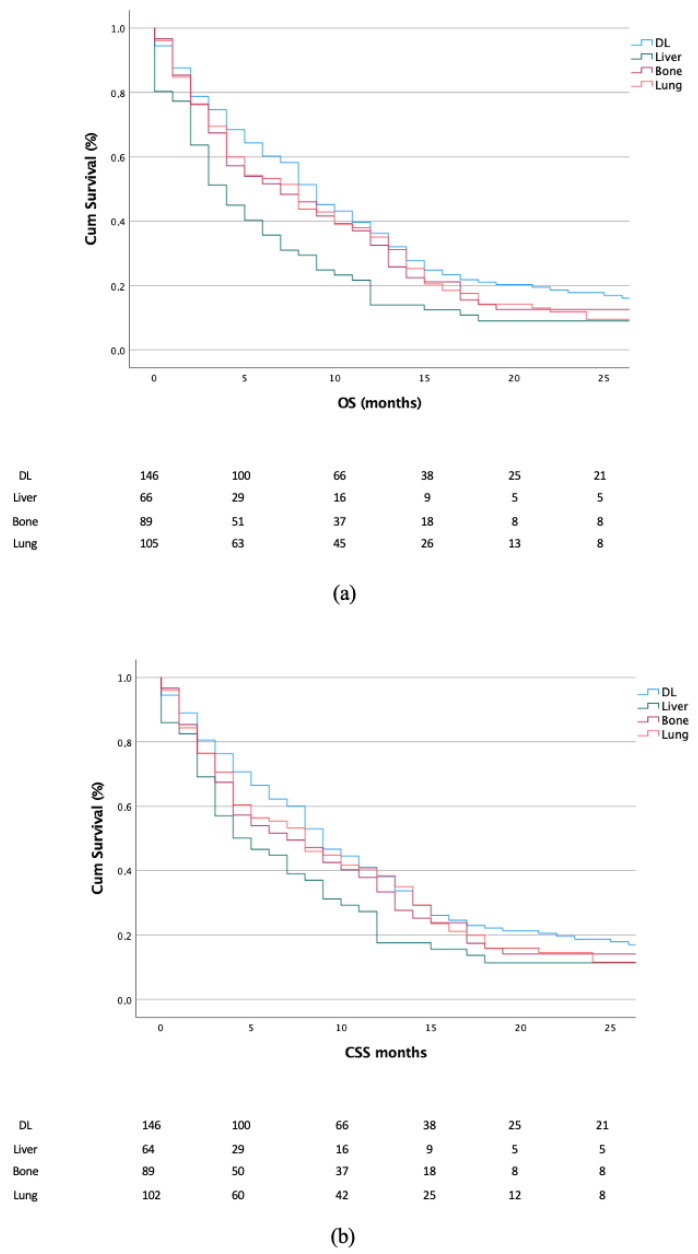
Kaplan–Meier plots depicting OS (**a**) and CSS (**b**) according to different metastatic organs in patients presenting single-organ metastases. DL, distant lymph node.

**Figure 3 jcm-11-05310-f003:**
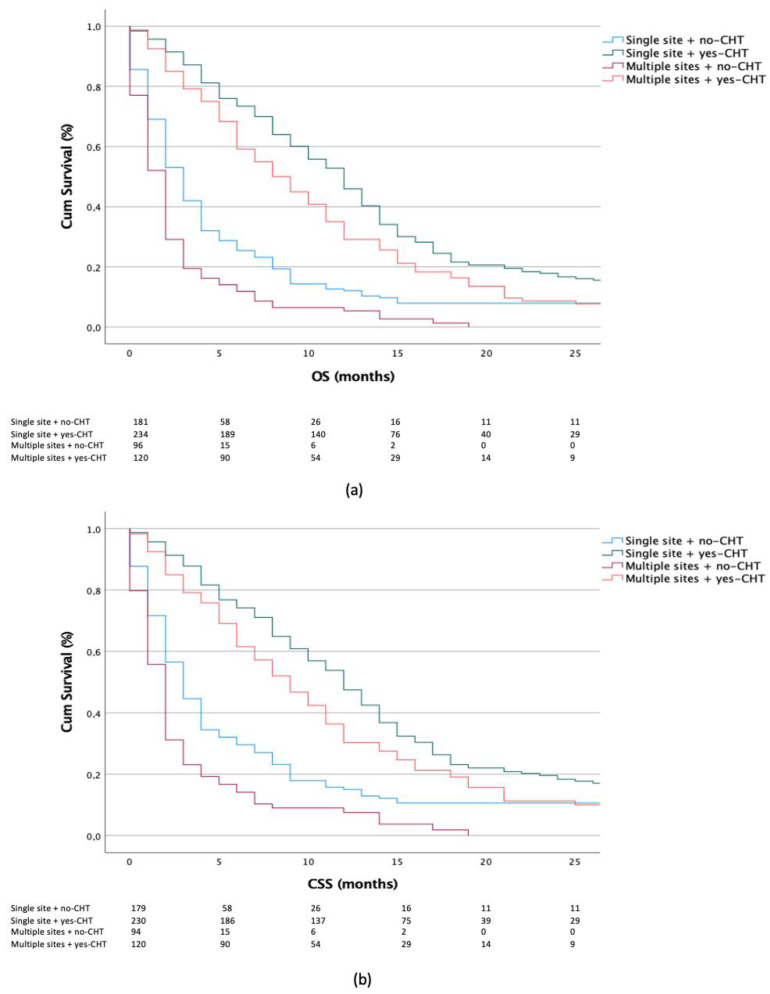
Kaplan–Meier plots depicting OS (**a**) and CSS (**b**) according to chemotherapy (CHT) administration in patients with single- vs. multiple-organ metastases.

**Figure 4 jcm-11-05310-f004:**
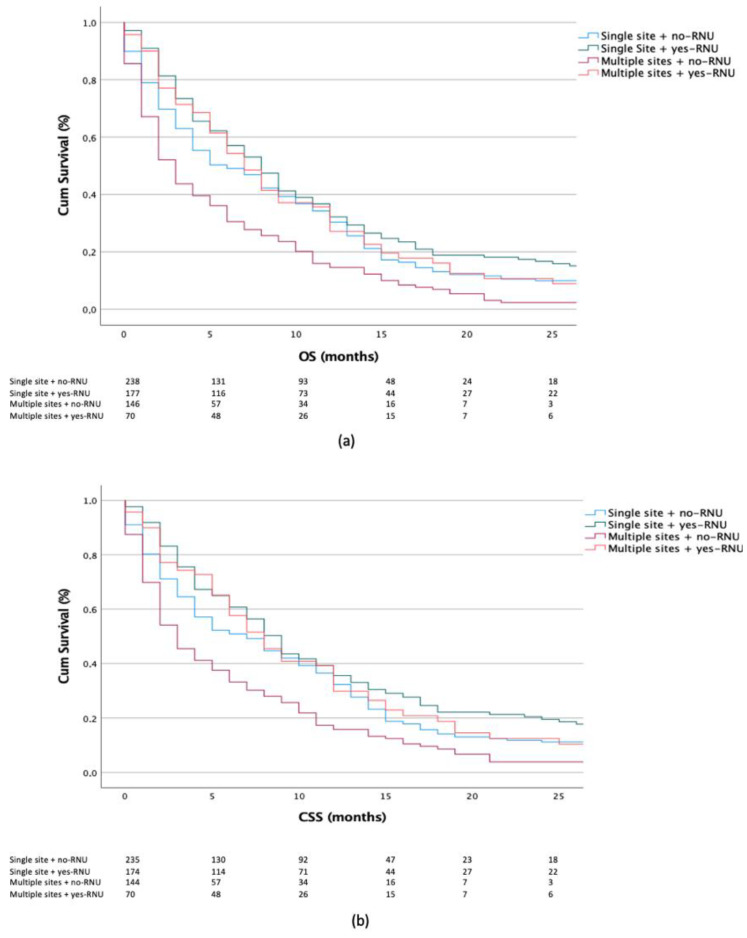
Kaplan–Meier plots depicting OS (**a**) and CSS (**b**) according to radical nephroureterectomy (RNU) treatment in patients with single- vs. multiple-organ metastases.

**Table 1 jcm-11-05310-t001:** Baseline characteristics of patients with metastatic UTUC.

Characteristics	No.	%
**Age (years)**		
<65	176	27.8
≥65	457	72.2
**Gender**		
Female	267	42
Male	366	57.8
**(T)NM**		
X	151	23.8
0–2	160	25.3
3–4	322	50.9
**T(N)M**		
0	202	31.9
>1	431	68.1
**Grade**		
Low	159	25.1
High	474	74.9
**Race**		
White	529	83.6
Black	38	6.0
Other	66	10.4
**Radiation**		
No	54	8.5
Yes	122	19.3
Unknown	457	72.2
**Radical nephroureterectomy**		
No	386	61.0
Yes	247	39.0
**Chemotherapy**		
No	195	30.8
Yes	354	55.9
Unknown	84	13.3
**Primary site**		
Renal pelvis	427	67.5
Ureter	166	26.2
Ureteric orifice	40	6.3
**Number of metastatic sites**		
Single organ site	417	65.9
Multiple organ sites	216	34.1

**Table 2 jcm-11-05310-t002:** Multivariate Cox regression analysis of factors that influenced OS and CSS outcomes.

*Factors*	Median OS (Months) HR (95% CI)	*p*-Value	Median CSS (Months) HR (95% CI)	*p*-Value
** *Age (years)* **				
*<65*	Reference			
*≥65*	1.196 (0.903–1.584)	0.213	1.167 (0.871–1.564)	0.301
** *Gender* **				
*Female*	Reference			
*Male*	0.900 (0.705–1.148)	0.394	0.946 (0.732–1.221)	0.946
** *Tumor location* **				
*Renal pelvis*	Reference			
*Ureter*	0.782 (0.592–1.034)	0.085	0.715 (0.530–0.965)	0.028
*Ureter orifice*	1.044 (0.656–1.662)	0.855	1.023 (0.630–1.662)	0.926
** *(T)NM* **				
*1–2*	Reference			
*3–4*	1.16 (0.869–1.553)	0.312	0.946 (0.732–1.221)	0.668
** *T(N)M* **				
*0*	Reference			
*≥1*	0.962 (0.733–1.262)	0.777	1.003 (0.753–1.337)	0.983
** *Number of organ metastasis* **				
*Single organ site*	Reference			
*Multiple organ sites*	1.425 (1.159–1.753)	<0.001	1.417 (1.141–1.760)	0.002
** *Radical nephroureterectomy* **				
*No*	Reference			
*Yes*	0.675 (0.514–0.886)	0.005	0.671 (0.505–0.891)	0.006
** *Distant metastasis* **				
*Lymph node*	Reference			
*Liver*	1.732 (1.234–2.430)	0.001	1.531 (1.062–2.207)	0.022
*Bone*	1.188 (0.849–1.663)	0.315	1.219 (0.863–1.721)	0.261
*Lung*	1.179 (0.861–1.615)	0.304	1.138 (0.817–1.585)	0.444
** *Chemotherapy* **				
*No*	Reference			
*Yes*	0.405 (0.313–0.523)	<0.001	0.435 (0.333–0.570)	<0.001

## Data Availability

https://seer.cancer.gov, accessed on 1 July 2022.

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
