# Peer review of "Prognostic Significance of Organ-Specific Metastases in Patients with Metastatic Upper Tract Urothelial Carcinoma"

_jcm, 2022, doi:10.3390/jcm11185310_

Round 1

Reviewer 1 Report (Previous Reviewer 1)

The raised comments have been all adequately addressed. I am glad that the Authors have considered our comments and I believe that the quality of the manuscript has been now improved. Happy to recommend acceptance of the manuscript in its present form.

Reviewer 2 Report (Previous Reviewer 2)

Good revisions. While I understand the authors' explanation of "UTUC with distant metastasis as the first primary malignancy" in their response, I feel it may still be confusing for the reader and should be clarified in the text, i.e. "UTUC with synchronous metastasis at presentation" would be clearer than calling it "the first primary malignancy"

This manuscript is a resubmission of an earlier submission. The following is a list of the peer review reports and author responses from that submission.

Round 1

Reviewer 1 Report

In the Manuscript by Antonio Tufano and colleagues, the authors attempt to investigate the prognostic associations of the metastatic site with the outcome of metastatic upper tract urothelial cancer (mUTUC). Although the topic is very relevant and timely, there are statistical shortcomings in the proposed methodology and several aspects that should be clarified before the article can be found acceptable for publication.

Major comments:

1. It is not clear why the authors selected to focus only on the single- organ metastatic sites? Please provide the rationale behind the study design, even more considering that number of metastatic sites is a significant predictor or overall (OS) and cancer specific survival (CSS).

2. Based on the multivariate cox regression analysis, apart from the metastatic site, significant prognostic value (even with lower p values reported), is also shown according to the followed treatment both for chemotherapy and radical nephroureterectomy. This is very important, and it is later on overlooked when plotting by using the Kaplan Meier curves.  It seems that any significant observation regarding the association of the metastatic site and survival outcomes is probably attributed mostly to the treatment scheme. Please provide stratified survival analysis to account for the treatment benefits.

3. It is unclear why the authors did not include radiation treatment in the multivariate cox regression analysis.

4. Please separate the unknown variable entries as a distinct group (eg. for radiation and chemotherapy). They are currently pooled with the “No treatment” group.

5. Please remove the marital status from the table with the baseline characteristics. This is not involved in the statistical analysis. Since this was not included in the statistical analysis, thus was considered irrelevant to the study, the information should be omitted.

6. Please include the number of events in the Kaplan Meier plots.

7. Please include a table (flowchart) to describe how many patients had to be excluded according to the different inclusion/exclusion criteria.

8. The references, particularly regarding the EAU guidelines and the epidemiological/ demographic parts in the introduction are outdated. Please consider the most recent guideline papers.

 Minor:

There seems to be a syntax error when citing the references 4,5,6 in line 51

Reviewer 2 Report

Overall this is an interesting study providing insight in the natural history of upper tract UC, and is worth publishing given the limited data currently available in this setting. However, there are some specific points that the authors should address:

1 - What is meant by "UTUC with distant metastasis as the first primary malignancy" (page 2, line 65). Was the SEER database queried strictly for patients with synchronous primary and metastatic UTUC, or are patients with metachronous metastatic disease included? The rate of RNU would seem to indicate that many patients had metachronous disease. If so, the authors should present this data; otherwise it appears that a large number of patients may have undergone RNU in the metastatic setting?

2 - The comparison with the Tanaka paper is somewhat inaccurate, as these authors looked at a cohort of patients who all underwent RNU (presumably largely for localized disease) .

3 - The paragraph focusing on chemotherapy administration would benefit from being rewritten. There is a mix of data around perioperative and palliative chemotherapy, and it is difficult to understand which better applies to the author's cohort. It would also be interesting to know if there was an interaction between chemotherapy use and sites of metastasis, as there is likely a selection bias where patients with more aggressive disease may not be fit for systemic therapy.

Minor point: on page 7, line 146 "metastasis" should likely be changed to "primary".